# Experimental Study on Hydrothermal Polymerization Catalytic Process Effect of Various Biomass through a Pilot Plant

**Alexis F. Mackintosh** [1], **Hyunchol Jung** [2], **In-Kook Kang** [3], **Seongyeun Yoo** [3], **Sanggyu Kim** [4] **and Kangil Choe** [5,*]

[1] PCS Technologies Inc., #204-3800 Westbrook Mall, UBC, Vancouver, BC V6S 2L9, Canada; alexis.mackintosh@gmail.com

[2] Department of Dental Biomaterials, School of Dentistry, Jeonbuk National University, 567 Baekje-daero, Deokjin-gu, Jeonju-si 54896, Jeollabuk, Korea; jhc0699@daum.net

[3] Bioenergy Center, Kinava Co., Ltd., #701-704 7 Heolleung-ro, Seocho-gu, Seoul 06792, Korea; ikkang@kinava.com (I.-K.K.); syyoo@kinava.com (S.Y.)

[4] Construction Division, Korea East-West Power Co., Ltd., #395 Jongga-ro, Jung-gu, Ulsan 44543, Korea; sgkim@ewp.co.kr

[5] Department of Mechanical Engineering, Hanyang University ERICA Campus, 55 Hanyangdaehak-ro, Bldg 5, Sangnok-gu, Ansan 15588, Gyeonggi, Korea

[*] Correspondence: kc15@caa.columbia.edu; Tel.: +82-10-9257-7851

**Abstract:** Through the previous study a hydrothermal polymerization (HTP)—a catalytic methodology for treating various biomass and organic wastes—has been developed on a lab scale with a 1 L reactor and the results published. The research work described herein aims to ensure that the catalytic process is scalable for pilot and even commercial scale plants. A 1700 L binary reactor system has been built and the assumptions of a commercial scale plant that would have 10,000 to 20,000 L pressure vessels tested. The HTP catalytic biofuel process converts mono- and polysaccharides into a solid polymer fuel that is based on a furfuraldehyde ring system. The calorific value of the material obtained from the pilot plant is on the order of 27 MJ/kg and the material typically has low ash and fixed carbon content order of 48% which are about same as the lab results for various wood biomass feedstocks. Though a 1700 times scale up binary reactor system the scalability of the HTP catalytic methodology has been confirmed and the mass and energy balance of the binary reactor identified in order to provide fundamental data for commercial scale establishment in future.

**Keywords:** hydrothermal carbonization; HTP catalytic process; pilot scale; biofuel; organic waste



## 1. Introduction

Green waste-to-energy (G2E) technology mostly deals with organic waste such as garden waste, food waste produced from municipality. Thanks to zero organic waste policy of many governments in Asia, North America and Europe, the G2E technology has been a hot subject in the waste treatment industry and waste-to-energy R&D community as well. Among the G2E technology three most popular subjects have been developed last decades: gasification, torrefaction, and hydrothermal carbonization (HTC) [1,2]. Unlike gasification, torrefaction and HTC technologies may produce solid biofuel through a thermal-chemical process at a comparatively low temperature about 150 to 350 °C [3]. The distinction between these two processes is that the HTC reaction is a wet process using water as a solvent while the torrefaction is a dry process requiring more energy. The HTC usually requires shorter reaction times (1 to 12 h) at a relatively lower temperature range (180~250 °C), with corresponding pressures up to 3 MPa [4,5]. It has been proved that the HTC process works well not only with biomass, but also with various feedstocks such as the organic part of MSW, paper, food waste, and animal manure [6–9].

The reaction conditions of HTC can be effectively reduced by adding catalysts. Shimizu et al. used various solid acid catalysts such as heteropolyacids, zeolites, and acidic resins [10]. They showed that the catalyst increases favorably the 5-hydroxymethylfurfural (HMF)

yield in fructose dehydration while it decreases the undesired hydrolysis of HMF to levulinic acid. Lynam et al. reported the effect of acetic acid and lithium chloride for the HTC reaction of lignocellulosic biomass [11]. They demonstrated a 30% increase of higher heating value (HHV) comparing to a non-catalytic process. Mackintosh et al. compared various study results on the catalyst effect on HTC with newly defined efficiency of the catalyst on HTC [12]. They used maleic acid as catalyst and optimized the operating conditions for wood chips: temperature = 220 °C, pressure = 2.3 MPa, process time = 1 h, and the amount of the catalyst = 20 g/L. Their so-called hydrothermal polymerization (HTP) process attained a high energy density of 27 MJ/kg and a mass yield rate of 60%. By using catalysts, the process temperature was lowered by 10 to 40 °C, the pressure requirement was reduced by 1 to 2 MPa, the rate of yield was 22% higher, and the total processing time was shortened by 3 h. These facts tell us the HTP process can use milder and quicker operation conditions with higher energy value and yield rate comparing with other catalytic and non-catalytic HTC reactions for wood biomass [7,12–14].

For commercialization some scale-up studies of the HTC process have been attempted. Hoekman et al. developed a scaled-up HTC system (by a factor of 20) in order to process 3 kg of lignocellulosic biomass and showed that the results were in good agreement with their lab scale 2 L reactor ones [15]. Owsianiak et al. assessed the environmental performance of commercial scale HTC reaction system in 15 categories for four wet biomass waste streams: green waste (garden waste), food waste, organic part of municipal solid waste, and digestate from anaerobic digestion for agricultural waste [16]. As a replacement for traditional waste treatment system, they suggested that the hydrochar from HTC of these four different biowastes can be an attractive option with an energy recovery system. Ismail et al. reported a numerical assessment for HTC for the conversion of municipal waste into hydrochar at a commercial scale [17]. They used commercial code with development of a transient model facilitated the calculation of important parameters such as operation temperature, pressure, gas flow and composition respect to process time. Eventually they recommended the HTC treatment system as an alternative to current MSW treatment plants.

Despite the many trials for commercial scale HTC processes there has been no published scale-up study of catalytic HTC processes so far. As well known in catalyst studies, the scalability of the catalyst is always a critical issue. Therefore, based on the previous lab results of HTC with the hydrothermal polymerization (HTP) catalyst reported by Mackintosh et al. [12], a pilot plant has been established for a scalability check-up and future commercial plant design. In this study, the 1700 L binary reactor system has been designed for the sake of energy efficiency. The applied binary reactor system is currently being used in various chemical synthesis processes such as the ethylene polymerization process [18] and paper mills as well.

## 2. Experimental Setup and Binary Reactor System

A binary reactor system was implemented to reduce the energy requirement of the HTP process by passing steam from the reactor that has finished processing the feedstock to a reactor that is ready to be heated. The steam transfer accomplishes two things. First the steam transfer from the reactor heats the cold reactor while cooling the hot reactor thereby transferring a significant amount of energy that would otherwise be wasted. Secondly, volatiles in the hot reactor are transferred to the cool reactor for further processing; the volatiles transfer will also transfer a significant amount of catalyst thereby reusing the catalyst in the next batch.

The reactors were heated using live steam injection into the bottom ports of the reactor from a 300 KW electric boiler system. The boiler capacity was 426 kg/h of steam. The maximum pressure was 40 kg/cm$^2$ while the operating pressure was 35 kg/cm$^2$. The boiler was first heated to about 240 °C. The controller on the boiler regulating the pressure in the boiler reservoir was set to 35 kg/cm$^2$ which is the saturated steam pressure of hot compressed water at 241 °C [19].

Once the boiler was up to pressure the valve to the reactor system was opened and the live steam was injected into the bottom of the reactor causing agitation and heating of the biomass. When the pressure of the reactor was equal to that in the boiler a check valve prevented back flow of the reactor contents into the boiler. The temperature in the reactor was maintained by periodic addition of steam from the boiler (Figure 1).

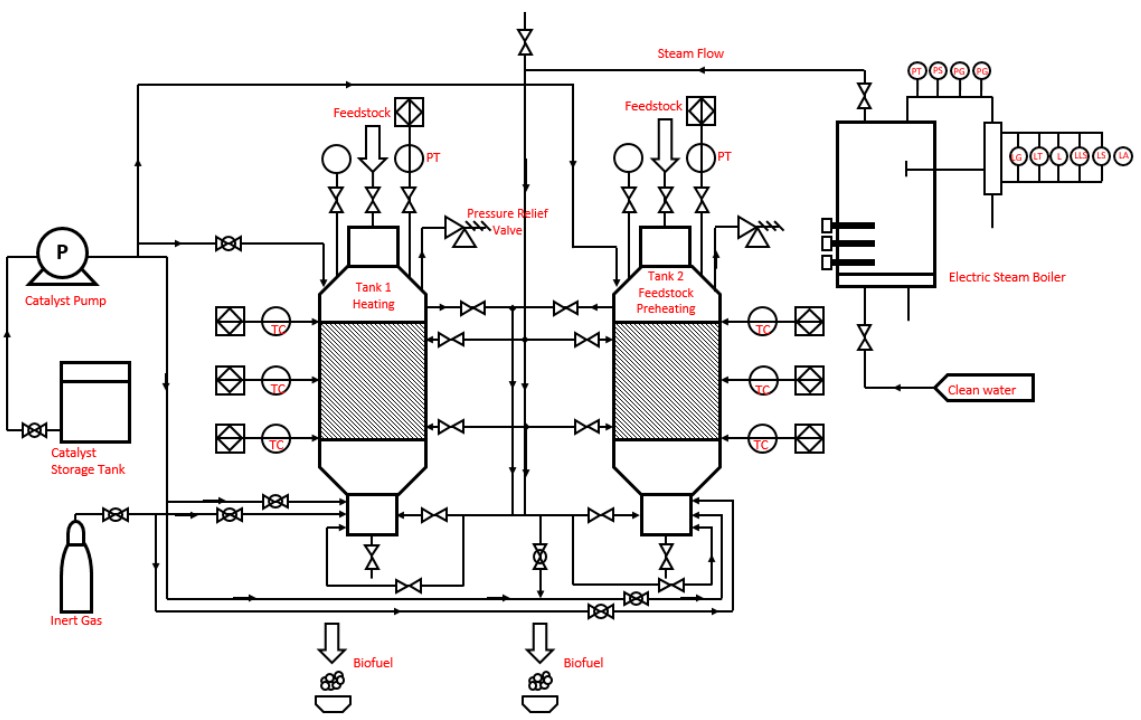

**Figure 1.** Layout of experimental setup and instrumentation for the HTC binary reactor (1700 L × 2) and boiler.

During testing of the reactor system a standard procedure was instituted for the production of biofuel. The test results reported below for several feedstocks such as sawdust, bark and chip, and palm waste were produced using the following procedure:

- 200 kg of feedstock (as received) was placed in the reactor pressure vessel.
- 200 kg of catalyst solution was added to the reactor system.
- The reactor was sealed and live steam was injected into the bottom of the reactor to heat the biomass/catalyst solution to between 230 °C and 240 °C. This process took about 1.5 h and added about 300 kg of water to the system.
- The reactor was held at this temperature for 2 h.
- After the process time the reactor was cooled by releasing the steam from the reactor. Once the reactor was at atmospheric pressure and below 100 °C the bottom valve was opened slightly and the catalyst solution allowed to drain out.
- Finally the bottom ball valve was opened and the resulting biofuel was recovered. Samples were collected and sent to a lab for analysis.

### 3. Thermal Analysis of the Binary Reactor System

For the HTP hydrothermal process to be economically feasible, the rate of heating and cooling of the reactor must be rapid. That is, on the order of under 1 h to heat the tank to temperature. The best method of heating the tank to the required temperature and pressure is to inject live steam into the bottom of the reactor. This steam will condense releasing its heat of evaporation into the tank thus heating it rapidly. While the engineering of live steam injection is well known [20], the effect of it on the biomass conversion to biofuel process under the subcritical condition is not certain and must be investigated prior to a commitment to the use of live steam for heating the reactors.

### 3.1. Heat Transfer between Reactors

The binary pair will operate by having one tank at temperature and pressure while the other tank is being loaded and prepared for processing. Once the first tank is finished its cook, the steam off the top of the reactor will be transferred to the bottom of the second reactor to recapture the heat from the first tank thereby heating the reactor. This process was tested using a bark feedstock. The major parameters are the pressure and temperature differentials. Also, the size of the piping between the two reactors needs to be of such size to prevent flashover while also allowing equilibrium between the two tanks in a 15 min period.

Figure 2 below plots the pressure and temperature in the two reactors as a function of time. The time frame covers the entire cycle of the binary reactor system. The boiler system regulated pressure in the reactors so the pressure remains constant once the reactor is up to process temperature. When the left reactor's processing time is complete the valve to the boiler is closed and the valve to the right reactor was opened allowing steam to flow from the top of the left reactor to the bottom of the right reactor. The steam flow was driven by the pressure gradient, once the pressures became equal the flow of steam between the reactors stopped and equilibrium was achieved. The equilibrium pressure was noted to be 10 kg/cm$^2$ while the temperature was 179 °C at the bottom of the reactor. The difference between the top and the bottom of the right reactor is due to the thermal mass of the reactor shell taking longer to heat to equilibrium. The saturated steam pressure of 10 kg/cm$^2$ would indicate a temperature of 179 °C.

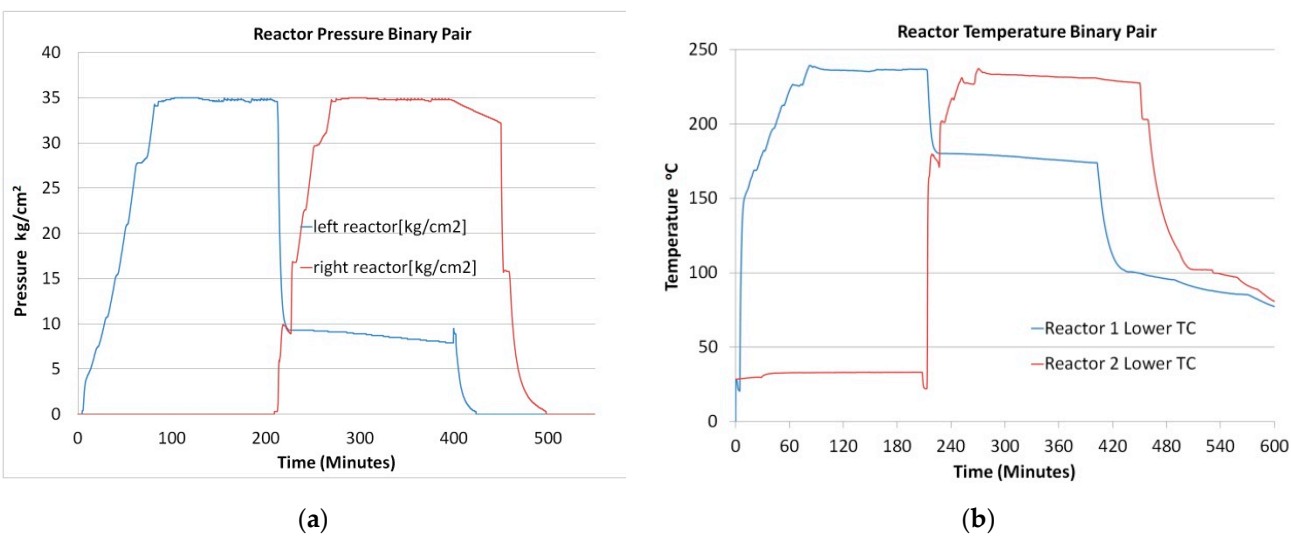

**Figure 2.** Pressure profile (**a**) and temperature profile (**b**) of the binary reactor pair over entire heating cycle. The boiler was set to 35 kg/cm$^2$ and pressure was regulated.

As shown in Figure 2 the temperature in the left reactor drops as the processing time increases. This temperature decrease is due to the evolution of gasses from the feedstock. Since the feedstock contains volatiles (woody biomass can be up to 3–5% volatiles), the rapid heating of the biomass will result in the volatiles boiling off. Since the reactors are regulated by temperature the gas evolution manifests itself as a decrease in temperature. After the initial decline there is a further slower decline due to the formation of $CO_2$ during the conversion process. It is interesting to note that the $CO_2$ mediated decline is larger in the second reactor than in the first cook. This would indicate that $CO_2$ formed in the first reactor is transferred to the second reactor thus suppressing the operating temperature when the system is regulated by pressure. A method to reduce this effect would be to leave the second reactor open to the atmosphere at the beginning of the transfer process thereby allowing the $CO_2$ to escape.

Figure 3 gives detailed view of the pressure and temperature changes that occur in the two reactors when they are connected. The steam from the top of the left reactor flows to the bottom of the right reactor where it condenses heating the catalyst and biomass in the right reactor. The thermal transfer is quick; essentially complete in about 5 min. The step in the pressure and temperature curve around 214 min is probably due to the time the liquid in the reactor takes to boil. We found that there was a crust buildup on the surface of the catalyst solution from components in solution polymerizing at the surface of the liquid (Figure 4). The formation of solids on surface is expected to reduce the rate of steam production.

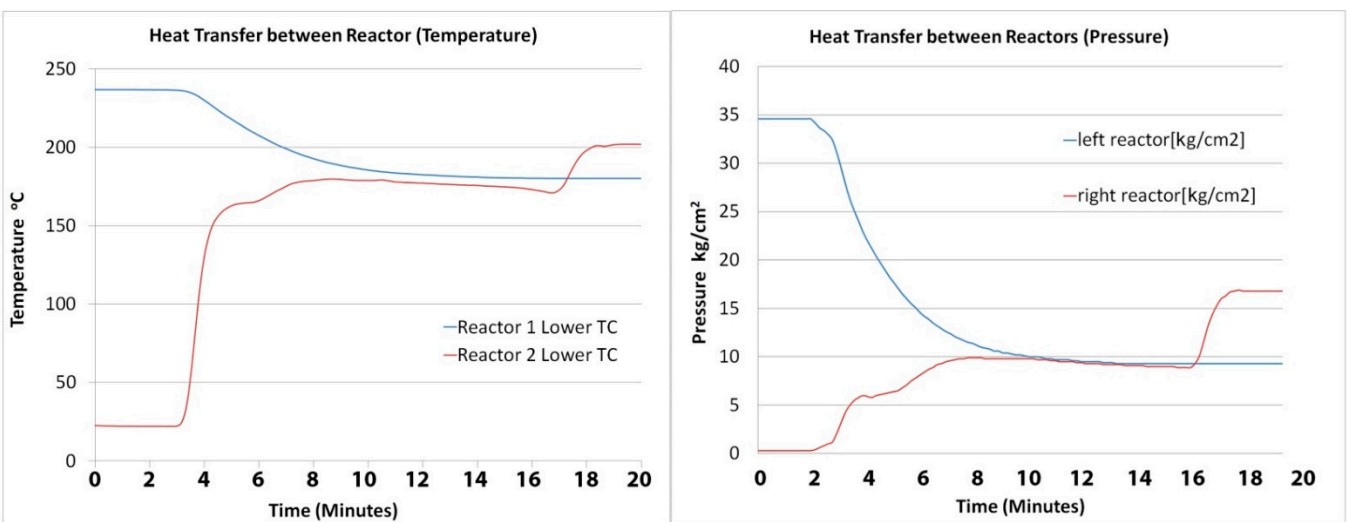

**Figure 3.** Temperature and Pressure profile (detail) of the binary reactor pair during heat transfer. The boiler was off until 226 min where the valve to right reactor was opened.

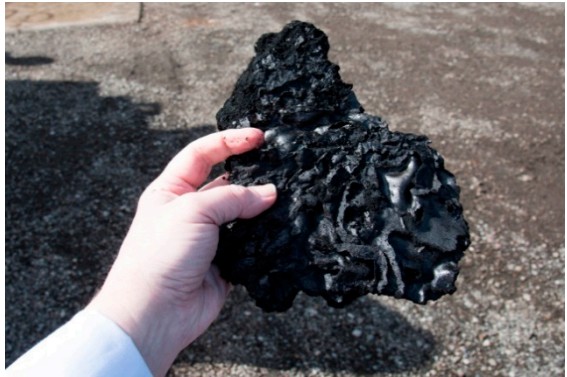

**Figure 4.** Photograph of crust from top of biofuel slurry after evaporative cooling of reactor.

In the reactor that is cooling the temperature drops fairly smoothly and equally over the entire volume of the reactor as illustrated by Figure 5 which plots the temperature of the top, middle and bottom of the reactor. Interestingly, the same cannot be said for the heating of the right reactor. The lower thermocouple heats very quickly and to a higher temperature than does the middle and upper thermocouples. Due to the loading of the reactor only the bottom thermocouple is in the solution while the middle and top thermocouples are above the liquid. Most of the steam being injected into the reactor system would be expected to condense causing the liquid to heat faster than the material above. It had been assumed the injected steam would be sufficient to mix the biomass/catalyst solution to prevent this temperature stratification. Clearly, this assumption is incorrect. To address the issue the

amount of catalyst solution needs to be increased so that the slurry will mix better and/or a stirrer needs to be added to the system.

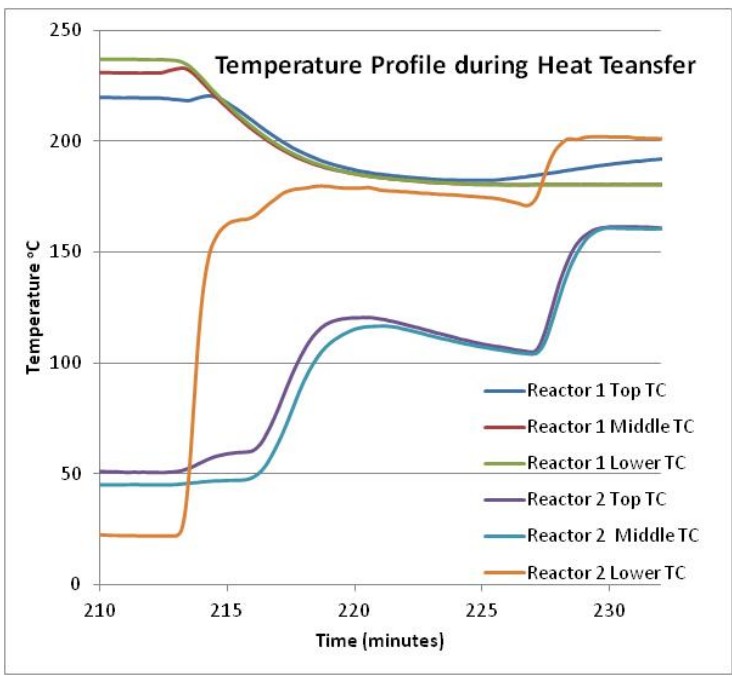

**Figure 5.** Temperature profile of all six thermocouples in the left and the right reactors during heat transfer process.

### 3.2. Mass and Energy Balance of Binary Reactors

Based on the heat transfer analysis above, mass and energy balance of the binary reactors was determined as seen below (Figure 6). Once the first reactor's cooking is done, it releases 116,000 kcal of energy. The second reactor retrieves 104,300 kcal of energy from the first reactor which is about 89.9% of the flash steam energy (116,000 kcal) and 63.1% of the total input energy (165,200 kcal) holding by the first reactor. The waste energy including 11,700 kcal of the flash steam and 49,200 kcal of the biofuel slurry output at 100 °C can be used as preheating source for feedstock through waste heat recovery system like a heat exchanger. The recycled energy is able to increase the temperature of the 2nd reactor to 179 °C so that it requires only 42,700 kcal of energy more out of 165,200 kcal total input requirement. This tells that about 74% of the input energy is saved through steam flash between the two reactors.

## 4. Results and Discussion

The bark and chip tested appeared to be the type of material that is used for landscaping. The feedstock was measured to have a moisture content of 38%. Thus, the dry weight of the 200 kg of wet feedstock used was 124 kg. Two hundred kilograms of wet bark and chips were placed in the bioreactor and 200 kg of catalyst concentrate was added. The catalyst used was the same maleic acid ($C_4H_4O_4$) developed through the previous study in the lab scale [12]. The concentration of the catalyst concentrate was 0.14 molar with a pH of 1.56.

### 4.1. Effect of Washing Biofuel from Bark and Chips

Samples of the biofuel produced from bark and chip were washed by placing the biofuel in a coarse #10 mesh (2 mm) fabric bag and immersing it in several changes of water. The wash water was black so the solids were allowed to settle out from the wash water and were collected for analysis. The proximate analysis determined by thermogravimetric analysis is done. As expected, the fines from washing had lower ash content.

The fines are less than 2 mm which is much smaller than the bulk of the biofuel so are more effectively washed removing both ash components and the soluble volatile components in the biofuel.

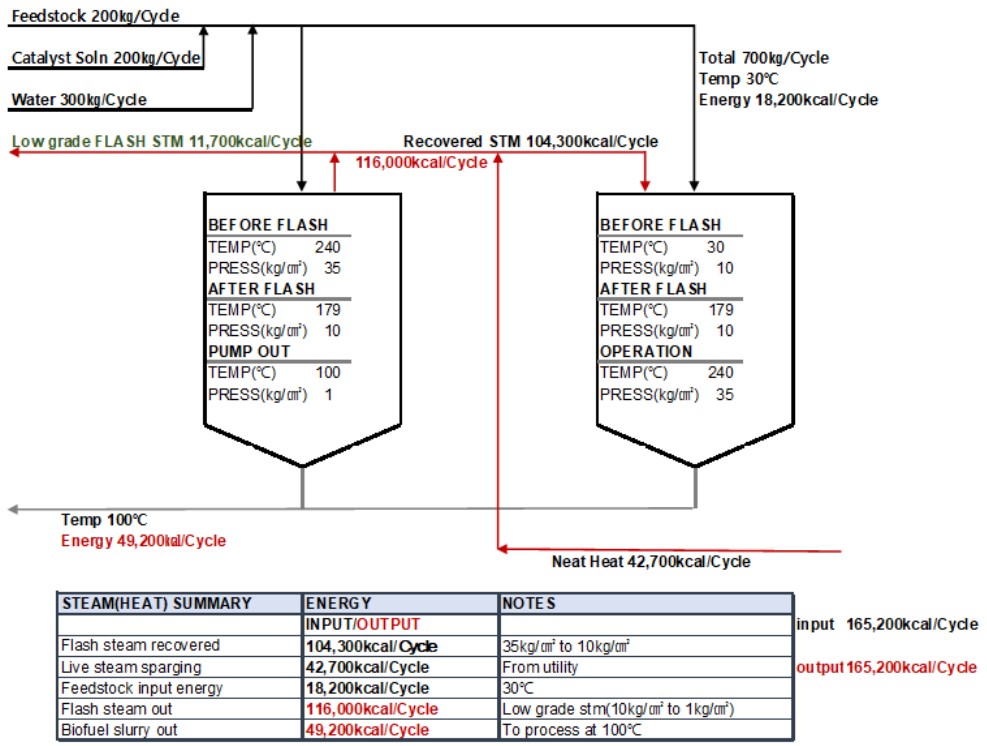

**Figure 6.** Mass and energy balance between the two reactors with bark feedstock HTC process.

The proximate analysis was performed on air dried material using a Mettler TGA/DSC (Mettler-Toledo, LLC, Columbus, OH, USA). The method was a modified ASTM E1131 method. The calorific value was determined using a PARR 6200 Isoperibol Bomb Calorimeter (Parr Instrument Company, Moline, IL, USA). The method was the standard method recommended by Parr Instruments Inc. The proximate analysis and calorific value (HHV) data is shown in Table 1 below.

**Table 1.** Proximate analysis and calorific value of the biofuel from bark and chips.

|  | Biofuel from Bark and Chip (<2 mm Fines Washed) | Biofuel from Bark and Chip (Unwashed) |
|---|---|---|
| Moisture | 2.83% | 2.28% |
| Volatiles | 49.67% | 49.30% |
| Fixed Carbon | 43.81% | 42.44% |
| Ash | 3.69% | 5.98% |
| Calorific Value (HHV) | 25.9 MJ/kg | 24.5 MJ/kg |

### 4.2. FTIR Analysis of Biofuel from Bark and Chips

Samples of the biofuel from bark chip biomass were analyzed using a 6700 mid-infrared Fourier transform infrared spectrometer (Thermo Nicolet, Waltham, MA, USA) equipped with KBr optics and a DTGS detector and a Smart iTR module with a diamond anvil. The biofuel spectrum was consistent with those we have collected from other woody-based biofuels. Figure 7 below shows the entire spectrum from 4000 cm$^{-1}$ to 600 cm$^{-1}$.

The FTIR spectral comparison (Figure 8) of the biofuel from bark shows the changes to the biofuel properties caused by washing. The two spectra were normalized to each other by matching the –OH peak between 3200 and 3400 cm$^{-1}$. The broad –OH peak is characteristic of alcohol functional groups in the biofuel. Comparing the absorbance of the

biofuel in the region of 800 cm$^{-1}$ and 1800 cm$^{-1}$ to the –OH region we note that absorbance of the alcohol groups in the washed sample is less than that in the unwashed sample. This clearly indicates that more soluble alcohol and carboxylic acids components were removed by washing.

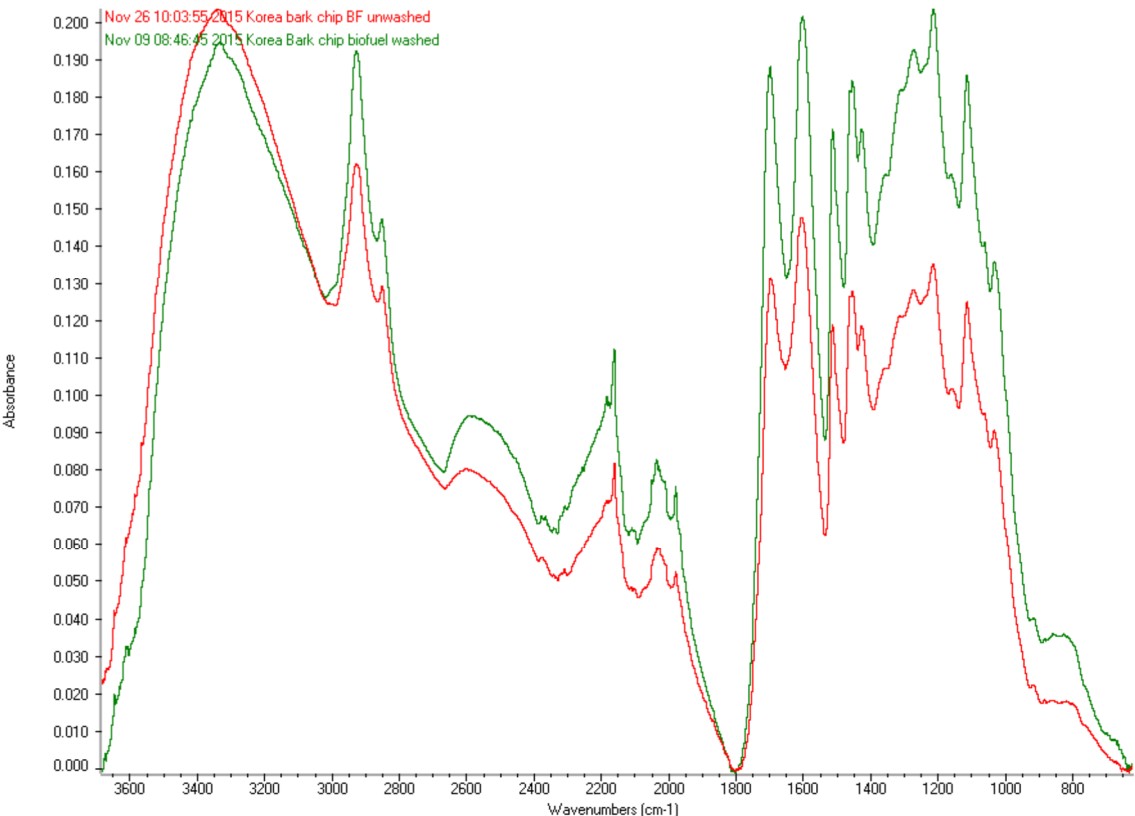

**Figure 7.** Fourier Transform Infrared Spectroscopy (ATR method) of the biofuel from bark and chip. Red line is biofuel as received, unwashed and green line is biofuel after being washed.

### 4.3. Biofuel from Sawdust Feedstock

The sawdust feedstock was a wet coarse woody material. The moisture content was measured to be 27.5% Since 200 kg of sawdust was loaded into the reactor we can calculate the dry weight equivalent of the biomass to be 145 kg. The catalyst concentrate volume was 200 kg. The catalyst type was maleic acid ($C_4H_4O_4$) catalyst dissolved in dilute phosphoric acid. The pH of the concentrate added into the reactor was 1.56. The biomass was placed in the reactor prior to putting in the catalyst solution resulting in incomplete mixing of the catalyst solution with the biomass. The consequence to this defect was that the bottom portion of the biomass was not fully processed. Samples analyzed herein are from the middle portion of the biofuel (Table 2).

**Table 2.** Proximate analysis of the biofuel from wood (sawdust), in washed and unwashed conditions.

|  | Biofuel from Sawdust (Washed) | Biofuel from Sawdust (Unwashed) |
| --- | --- | --- |
| Moisture | 3.0% | 2.8% |
| Volatiles | 46.6% | 47.5% |
| Fixed Carbon | 49.1% | 48.5% |
| Ash | 1.3% | 1.2% |
| Calorific Value (HHV) | 27.6 MJ/kg | 26.4 MJ/kg |

### 4.4. Infrared Spectroscopy of the Biofuel from Sawdust

Figures 8 and 9 are the Fourier transform infrared absorbance spectra of the biofuel produced in the 1700 L pilot plant reactor and the 1 L lab scale reactor. The spectra for the two samples are similar indicating that on a chemical functional group level the two materials are of similar product. There are some differences in the peaks at 1270 cm$^{-1}$, 1212 cm$^{-1}$ (ether C-O stretching) and 1172 cm$^{-1}$ ($-\nu$ C–O–C in cellulose and hemicelluloses) are likely due to the product from the large reactor having more unprocessed feedstock in the product than the 1 L batch. This fact provides strong evidence that the hydrothermal processes that we have observed in the laboratory are the same as those occurring in an industrial scale setting.

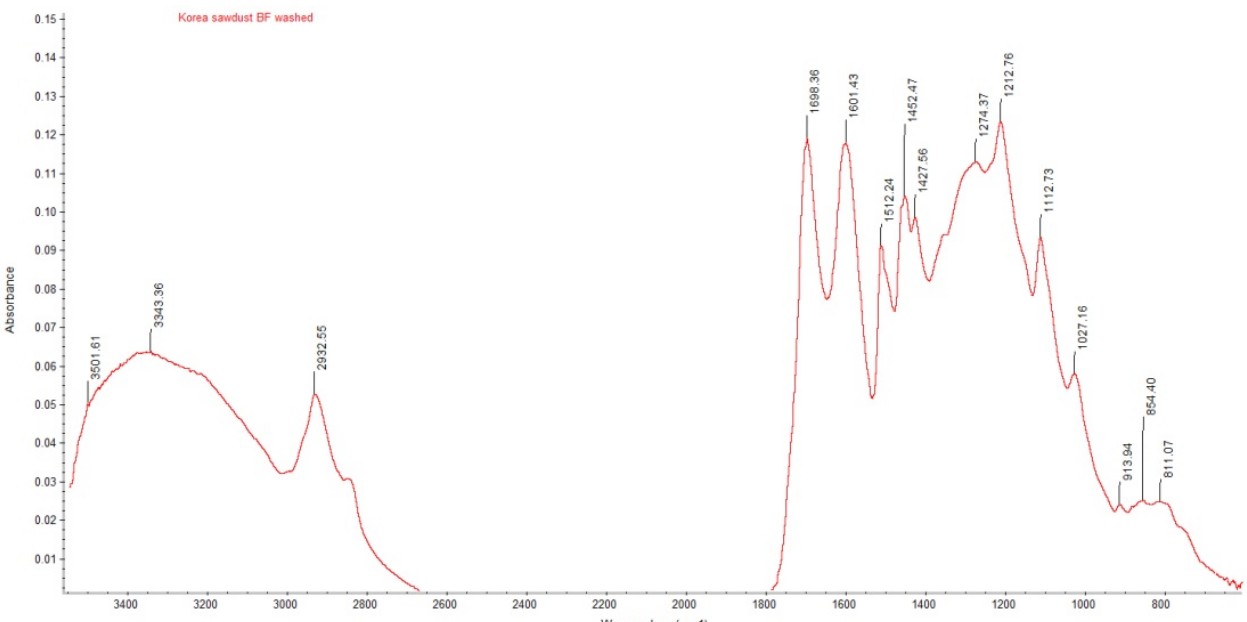

**Figure 8.** FTIR spectrum of the biofuel from sawdust produced from 1700 L reactor.

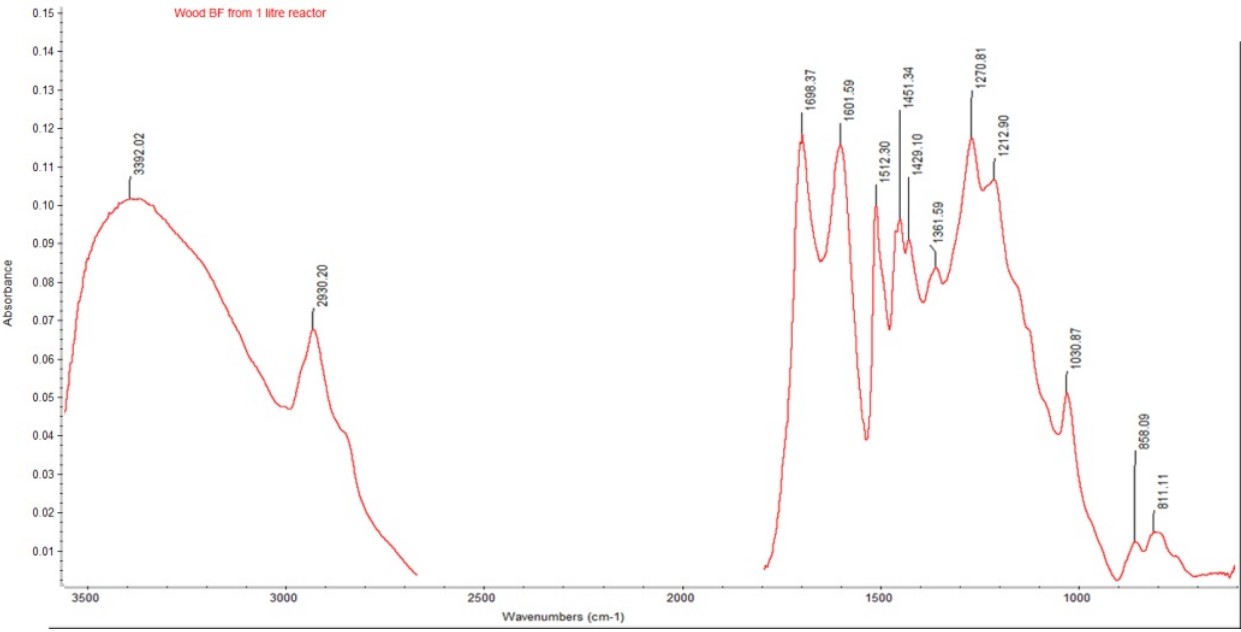

**Figure 9.** FTIR spectrum of the biofuel produced from softwood in 1 L lab reactor as reference sample.

### 4.5. Biofuel from Palm Oil Production Waste (Nut Husks)

The conversion of palm oil waste was processed using the standard method. The biomass (200 kg was added to the reactor followed by a maleic acid ($C_4H_4O_4$) catalyst concentrate. The input catalyst concentration was 1.56. After processing the catalyst recovered had a pH of 3.47. The change in pH is due to two factors: the inherent moisture in the biofuel and the water added to the reactor during the live steam injection process. The process time was 2 h and the process temperature was between 230 °C and 240 °C. A significant portion of the palm nut waste in the lower section of the reactor was not fully processed and led to varying quality of the biofuel produced. We sampled the material in two aliquots: the first being the light brown shells which were not fully processed and the black looking shells that were fully cooked. Samples of unprocessed palm kernel waste were not tested for analysis, but the proximate analysis of palm nut waste has been reported in the literature [21] and is presented in Table 3 below.

**Table 3.** Proximate analysis on oven dried palm kernel shell and palm fiber compared to biofuel produced with a 1700 L binary pair reactor.

|  | **Palm Kernel Shell** | **Palm Fiber** | **Biofuel from Palm Waste** |
|---|---|---|---|
| Moisture | 0% | 0% | 3.20% |
| Volatile matter | 74.60% | 74.59% | 47.35% |
| Fixed Carbon | 22.58% | 19.38% | 46.43% |
| Ash | 2.82% | 6.03% | 3.02% |
| Calorific Value (HHV) | 19.4 MJ/kg | 18.1 MJ/kg | 27.2 MJ/kg |

When we compare the volatiles and fixed carbon content from the unprocessed palm waste feedstock to that of the material converted to a biofuel, we note that there is a large decrease in the volatiles and a commensurate increase in the fixed carbon content. The increase in calorific value is significant in that the material once processed into a biofuel has a calorific value of a mid-rank coal and so will be an excellent replacement of fossil coal burnt in thermal power plants. It should also be noted that the partially processed material had a significantly lower volatile content (67.6%) and higher fixed carbon content (28.4%) than did the unprocessed material.

### 4.6. FTIR Analysis of Biofuel from Palm Kernel Shell

Samples of the biofuel from palm kernel shell waste and the ash from the biofuel were analyzed using Fourier transform infrared spectroscopy (Figures 10 and 11). The biofuel spectrum was consistent with that we have collected from other woody based biofuels. The ash from combustion showed a very strong peak that is consistent with phosphates and silicate-based clays.

One notable difference is the lower intensity of the 1600 $cm^{-1}$ peak as compared to the 1600 $cm^{-1}$ peak. Since the symmetric aryl peak is stronger, this would suggest a higher content of aromatics from lignin compounds in the biofuel from palm kernels than from white wood. The broad peak at 3366 $cm^{-1}$ is due to –OH groups in the biofuel while the peaks at 2934 $cm^{-1}$ and 2843 $cm^{-1}$ are from symmetric and asymmetric bending of methylene –$CH_2$ groups. 1698 $cm^{-1}$ conjugated carbonyl (C=O). This peak is lower than usual 1740–1720 $cm^{-1}$ due to internal hydrogen bonding which occurs in conjugated unsaturated aldehyde. 1600 $cm^{-1}$ and 1513 $cm^{-1}$ are due aryl ring stretching symmetric and asymmetric respectively. 1450 $cm^{-1}$ and 1427 $cm^{-1}$ C-H deformation, symmetric and asymmetric. 1269 $cm^{-1}$ and 1215 $cm^{-1}$ indicate Ether C-O stretching vibration. The peak at 1115 $cm^{-1}$ is -ν C–O–C in cellulose and hemicelluloses while the 1030 $cm^{-1}$ is –ν C–O in cellulose.

The infrared spectrum of the ash is indicative of both phosphates and silicates being present in the biofuel. The strong peak at 1027 $cm^{-1}$ is diagnostic for both phosphate and silicates while the peak at 795 $cm^{-1}$ is indicative of silicon dioxide. The peak at ~600 $cm^{-1}$ is ascribed to bending O-P-O vibrations indicating there are phosphates in the ash. [22]

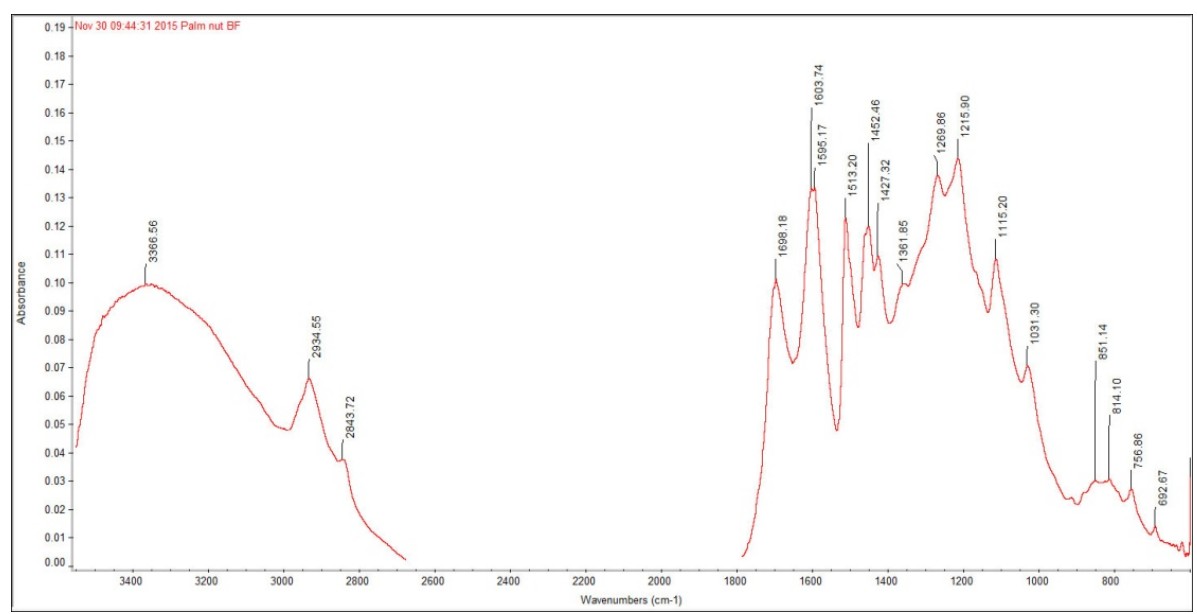

**Figure 10.** FTIR absorbance spectrum for the biofuel produced from palm waste.

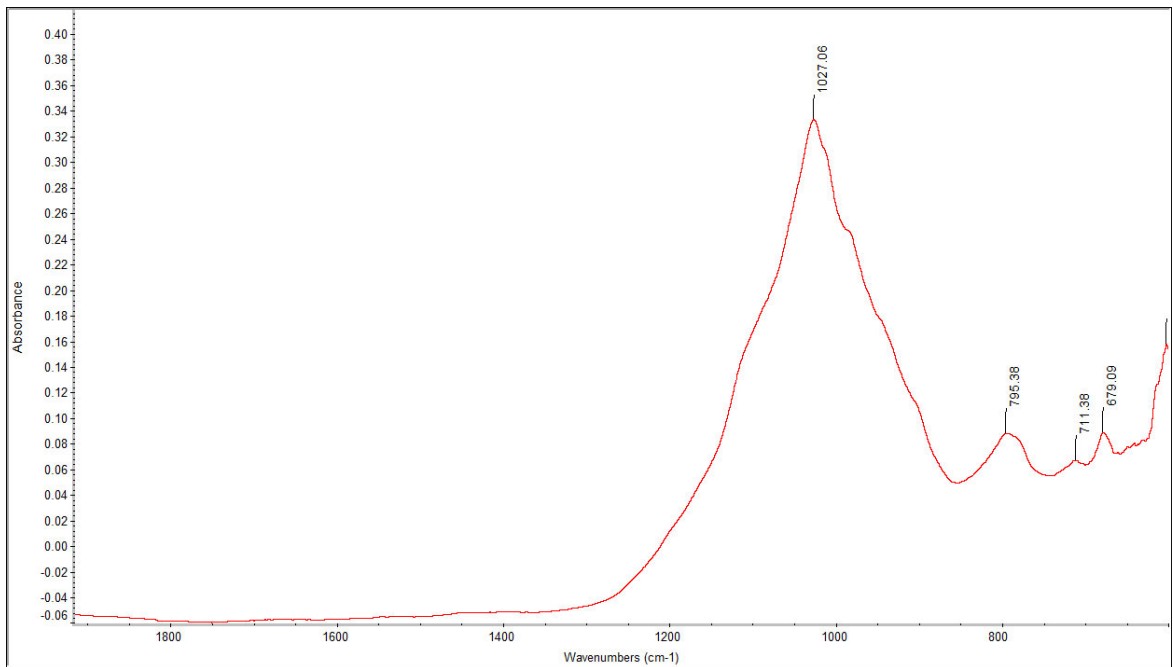

**Figure 11.** FTIR spectrum of ash from burnt biofuel from palm waste. The major peak in the spectrum is indicative of inorganic phosphates or silicates.

For van Krevelen diagram plot element analysis for C, H, O has been done and the atomic percentage ratio of O/C and H/C calculated (Table 4). In the van Krevelen diagram (Figure 12) the three biofuels made out of bark chips, palm waste, and sawdust are located near the zone of coal and lignite coal. With the lower O/C ratio and the higher H/C ratio, the fuel has a higher heating value. Therefore, it proved that the three biofuels from the pilot plant made from three different biomass feedstocks are well carbonized through the pilot plant and have a characteristic of regular coal in terms of the amount of carbon, hydrogen, and oxygen.

**Table 4.** Element analysis & comparison of O/C vs. H/C for three biofuels from Bark and Chip, Palm Waste, and Sawdust.

|  | C | H | O | O/C | H/C | Caloric Value (HHV) |
|---|---|---|---|---|---|---|
| Bark and Chip | 5.739 | 5.267 | 0.760 | 0.132 | 0.918 | 25.9 MJ/kg |
| Palm Waste | 5.574 | 4.871 | 1.571 | 0.282 | 0.874 | 27.2 MJ/kg |
| Sawdust | 5.495 | 4.762 | 1.623 | 0.295 | 0.867 | 27.6 MJ/kg |

(Unit: atomic %).

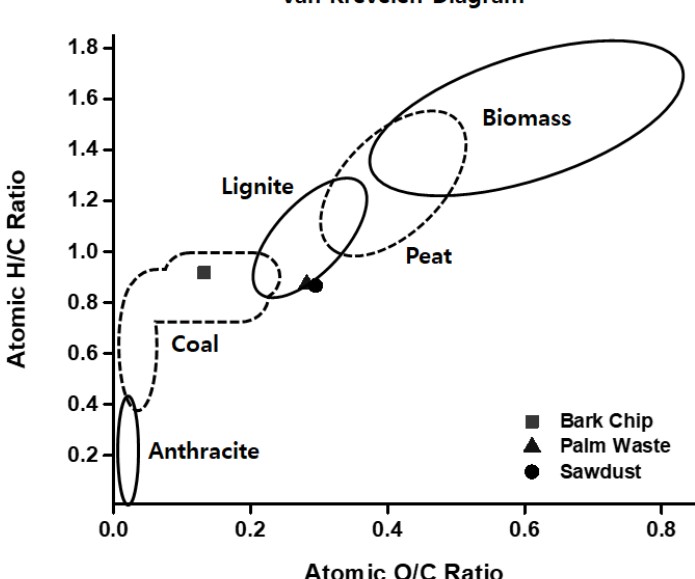

**Figure 12.** Van Krevelen diagram for three biofuels out of Bark and Chip, Palm Waste, and Sawdust.

## 5. Conclusions

Table 5 summarizes the results of the 1700 L pilot plant and compares with previous 1 L lab scale results [12].

**Table 5.** Proximate analysis and calorific value results. References for the lab results comes from previous work of Alexis et al.

|  | * Feedstock Wastewood | * Lab 1 L Wastewood | ** Lab 1 L Sawdust | 1700 L Sawdust | 1700 L Bark and Chip | 1700 L Palm Waste |
|---|---|---|---|---|---|---|
| Moisture (%) | 9.2 | 2.1 | 1.0 | 3.0 | 2.83 | 2.89 |
| Volatile Matter (%) | 72.3 | 56.8 | 50.2 | 46.6 | 49.67 | 47.35 |
| Fixed Carbon (%) | 17.1 | 39.8 | 48.4 | 49.1 | 43.81 | 46.43 |
| Ash (%) | 1.2 | 1.3 | 0.5 | 1.3 | 3.69 | 3.02 |
| Calorific Value: HHV (MJ/kg) | 17.8 | 26.1 | 27.0 | 27.6 | 25.9 | 27.2 |

* [12] and separate, ** lab results.

The above results demonstrated that the biofuel produced in the 1700 L reactor has the same properties as the biofuel produced in the laboratory using a 1 L reactor. This is also supported by FTIR spectroscopy results at Figures 8 and 9 and the van Krevelen diagram in Table 4 and Figure 12. Therefore, the results show conclusively that the biofuel process will scale to commercial size and that the results determined in the laboratory are the same as the results from a 1700-times larger reactor.

For building a commercial system, a mass and energy balance has been performed (Figure 6) through the study of heat transfer between two reactors. It showed that the flash steam movement between the two reactors enables a 74% energy savings. It provides a

tool to calculate the amount of steam that needs to be put into the system. However, it is regretfuk that the system didn't have a steam flow meter to double check the amount of the steam arising in each procedure.

The reactor system showed significant thermal stratification during runs with typical biomass feedstocks. The uneven temperature leads to a significant variability of the quality of the feedstock. In order to ensure thermal uniformity of the process a stirrer will be needed. Alternatively, a pump could be used to circulate the liquid from the bottom of the reactor to the top thereby heating the biomass more uniformly.

**Author Contributions:** The manuscript was written through contributions of all authors. These authors contributed individually such that A.F.M.: Writing-original draft preparation, H.J.: Conducting experiment and collect data, I.-K.K.: Analysis data and setup mass and energy balance, S.Y.: Writing—review & editing, S.K.: Supervise, K.C.: Writing—original draft preparation & funding acquisition. All authors have read and agreed to the published version of the manuscript.

**Funding:** This work was supported by the Korea East-West Power Company of the Republic of Korea (Pilot Plant Development for Green Pellet Production from Woodwaste Using Hydrothermal Polymerization Technology (2019)).

**Conflicts of Interest:** The authors declare no conflict of interest.

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
