# Peer review of "Experimental Study on Hydrothermal Polymerization Catalytic Process Effect of Various Biomass through a Pilot Plant"

_processes, doi:10.3390/pr9050758_

Round 1

Reviewer 1 Report

The manuscript entitled “Experimental Study on Hydrothermal Polymerization Catalytic Process Effect of Various Biomass through a Pilot Plant” reported a scaled up of the previous lab scale hydrothermal polymerization catalytic biofuel process that convert mono and polysaccharides to a solid polymer fuel. The energy consumption and efficiency throughout the process were calculated and compared. There are a few issues needs to be addressed.

  1. In Figure 8 and Figure 9, the range of x-axis was not set as the same with each other. Please consider revising the starting point and end point of Figure 9 to match with it as in Figure 8.
  2. The authors claimed that these two figures are identical, however, please explain the differences of the peaks in the wavelength range of 1000-1200 cm-1 in Figure 8 and 9.
  3. Please add references to the paragraph in line 343-344 on page 12 to support the claim of ~600 cm-1 ascribing to phosphates.

Author Response

Thank you for your comment and direction. Please check the detail answers for your comment as attached. On the manuscript I highlight the changes I made for your convenience. 

Reviewer 2 Report

Important and interesting topics. Well-motivated and utilitarian work. Calorific value of the material from the pilot station at the level of hard coal. One note: you have to distinguish between organic waste (waste biomass) and a by-product. This is because waste biomass is a valuable raw material in the energy production process, not a waste. Increasing the production capacity is indicated here, for example, in the production of furfulaldehyde (furfural), which has been almost unchanged for 100 years.

I have no objections to the presented HTC process. Drawings and charts legible.

In turn, the records of units require a large correction. In the SI system there is no such unit as Km, Kg or Kcal. Besides, there is no space (space) between the numerical value and the unit everywhere, this also applies to degrees Celsius, e.g. 350 °C. On the other hand, the tilde ~ is not a sign from ... to ... e.g. 1~12 hrs.

Author Response

(The authors gave the same response as above.)

Reviewer 3 Report

Review in the attachment 

Author Response

Thank you for your comment and direction. Please check the detail answers for your comment attached. On the manuscript I highlight the changes I made for your convenience.

Round 2

Reviewer 3 Report

The manuscript still contains some bugs. In my opinion, it is necessary to correct them before accepting the manuscript for publication. Details are included in the attachment "Review_2"

Author Response

Changes made:Authors should correct again the manuscript because:

  1. capital letters "Kg" - in the abstract have not been changed, and "Kcal" in lines 208, 209, 210, 211, 214, and “Kg” in the descriptions of the axes and legends of Fig. 2 and 3. --> fixed kcal to lower case k and fixed fig 2 and 3 to lower case "k" in kg
  2. Table 2 has been deleted and the sentence remained: "The infrared absorption peak assignment for the bark biofuel sample is listed below in Table 2."-->removed sentence as table has been removed.
  3. "Isoperibel" is also unchanged, although in response the Authors wrote that it was corrected. -->ops, misspelt isoberibol spelling corrected
  4. Please take care of spaces and superscripts: eg line 284 "1270cm-2841, 1212cm-1 (ether C-O stretching) and 1172cm-1" - it looks not good. -->Formatted to be superscript.
  5. Figures and Tables: line 494 "Bart"- it should be "Bark". -->corrected spelling in figures and tables.